# Barriers and facilitators for the implementation of Antimicrobial Stewardship Programs in Dar es Salaam Regional Referral Hospitals (RRHs)

Berthania Magesa[ID][1]*, Luco Mwelange[2], Nathanael Sirili[ID][1], Margareth Mhame[ID][2]

**1** Department of Development Studies, School of Public Health and Social Sciences, Muhimbili University of Health and Allied Sciences, Dar es Salaam, Tanzania, **2** Department of Environmental and Occupational Health, School of Public Health and Social Sciences, Muhimbili University of Health and Allied Sciences Dar es Salaam, Tanzania

\* berthymagesa03@gmail.com

## Abstract

Antimicrobial resistance (AMR) continues to present a significant public health challenge globally. Antimicrobial Stewardship Programs (ASPs) are key in optimizing antibiotic use and mitigating AMR's impact. This study explores the factors influencing the implementation of ASPs in three Dar es Salaam Regional Referral Hospitals (RRHs).We explored the facilitators and barriers for the implementation of ASPsin Dar es Salaam's RRH in Tanzania.Eleven purposively selected medical professionals from three RRHs participated in key informant interviews for this study, which employed an exploratory qualitative case study design. The audio recordings of the interviews were verbatim transcribed, and thematic analysis as proposed by Braun and Clarke (2006) was applied for anal-ysis.Two major themes were unveiled in this study: barriers to implementation of ASPs and the facilitators to implementation of ASPs. Resource constraints, staff shortage, inconsistency in adherence to guidelines, reluctance to change among staff, and logistical challenges in procurement of program consumables were highlighted as the barriers to implementing ASP. Leadership support, access to ASP resources through partnerships, and ASP-specific training and capacity-building initiatives were the major facilitators for implementing ASPs. The implementation of ASPs in Dar es salaam RRHs has established a foun-dational practice necessary to combat AMR. The study demonstrated the role of capacity building and institutional support in implementing these programs. Despite the benefits, these programs are challenged by systemic and behavioral limitations. While the programs show potential, their success remains fragile and dependent on addressing these critical, deeply ingrained systemic issues. Thus, the Ministry of Health should provide dedicated domestic funding and strengthen national governance structures to guide, coordinate, and monitor ASP imple-mentation across facilities, and at the hospital level, ASPs must be integrated

**Data availability statement:** Minimum data has been provided as part of the submitted article. (Please find the data as part of supporting documents labeled Transcript 1 & 2.).

**Funding:** The authors received no specific funding for this work.

**Competing interests:** The authors have declared that no competing interests exist.

with existing programs especially IPC with clear performance indicators, routine reporting, and targeted investments such as rapid diagnostics and standardized training to address operational and behavioral barriers.

## Introduction

ASPs are coordinated efforts to promote the appropriate use of antimicrobial medications, reduce the rising rates of AMR and decrease the spread of infections caused by resistant organisms to improve patients' outcomes [1]. The escalating threat of antimicrobial resistance (AMR) poses a significant challenge to global public health, necessitating urgent and effective interventions [2]. The World Health Organization (WHO) has warned that AMR threatens to render common infections increasingly difficult to treat, leading to increased morbidity and mortality; recent estimates that AMR attributes to over 1.27 million deaths annually. [3] This threat is particularly acute in Sub-Saharan Africa, where the widespread misuse of antibiotics and inadequate regulatory frameworks exacerbate the problem [3,4]. Across East Africa, AMR is influenced by various factors, including healthcare practices, agricultural methods, and socioeconomic conditions. The extent and nature of AMR in human and animal health vary significantly across the region, often correlating with local healthcare practices and agricultural use of antibiotics [5]. The prevalence of antibiotic use in Tanzania is at 62% with most people misusing or overusing antibiotics [6]. In a study done at a tertiary hospital in Tanzania showed that 47% of patients admitted are prescribed antibiotics empirically, which steers the burden of AMR [7]. National AMR data systems remains partially developed., Efforts like joining the East Africa Public Health Laboratory Network (EAPHLN) [8], conforming to WHO targets of AWaRE (Access, Watch, Reserve) categorization of Antibiotics [9], and complying and submit national antibiotics consumption and use data to Global Antimicrobial Resistance and Surveillance System (GLASS) indicate ongoing attempts to address and combat AMR in the country [10]. The pressing concerns are adherence to standard treatment guidelines, misuse of antibiotics, and the limited laboratory capacity for AMR monitoring, which severely hampers effective public health responses [11]. A review reported that only a small fraction of countries in the WHO African region have developed and implement national action plans (NAPs) to combat AMR, highlighting a significant gap in systematic surveillance and response mechanisms [12]. Public awareness and education surrounding AMR also play critical roles in its mitigation. Evidence points to a lack of understanding among healthcare providers and the general public about proper antimicrobial use and the implications of misuse [13].

ASPs are among the pivotal strategies introduced and designed to promote the appropriate use of antimicrobials and curb misuse [14]. They are typically multifaceted, involving education, adherence to prescribing guidelines, use of evidence-based protocols, and regular monitoring of antimicrobial usage and resistance trends [15]. Implementing ASPs has shown promise in various healthcare

contexts, demonstrating their ability to improve antibiotic prescribing practices, reduce healthcare-associated infections, and ultimately decrease the prevalence of resistant pathogens [16]. Prior efforts to address AMR targeted prescribers' and dispensers' behavioral change; ASP, on the other hand, utilizes a collaborative-persuasive approach of counselling and educating clinicians on the appropriate use of antimicrobials [17].

Antimicrobial stewardship interventions aim to prolong the effective life of antibiotics, and improve their utilization. [18]. However, the effectiveness of these programs can vary significantly based on local healthcare dynamics, including the availability of resources, the level of healthcare professional engagement, and the existing healthcare infrastructure [19,20]. Through the national action plan, Tanzania has been implementing ASP activities for five years with a focus on optimizing the use of antimicrobial agents in humans and animals, ensuring standard prescriptions, developing hospital formulary, auditing prescriptions, training healthcare personnel, and establishing therapeutic committees [10].

In the same context of Tanzania, the need for effective implementation of ASPs is underscored by the country's unique healthcare challenges, including limited access to diagnostic tools, inadequate training of healthcare personnel, and the pervasive use of broad-spectrum antibiotics without proper indications in humans and animals [21]. This has been evident with escalating resistance to commonly used antibiotics, including those for treating life-threatening infections such as pneumonia, sepsis, and tuberculosis [6]. A study showed that Enterobacteriaceae species isolated from hospital settings exhibited heightened resistance to common antibiotics, with notable resistance levels reported for co-trimoxazole (77.7%) and ampicillin (81.6%) [22].

ASPs implementation remains inconsistent across many low-resource settings, including Tanzania, where AMR continues to rise despite the National Action Plan on AMR launched in 2017 [21]. Previous studies have identified several barriers to the successful implementation of ASPs in similar settings, such as insufficient collaboration among healthcare providers, lack of institutional support, and cultural attitudes towards antibiotic use [23].

The implementation of ASPs in Tanzania presents a complex interplay of barriers and facilitators that require comprehensive understanding and strategic approaches. Addressing these challenges while capitalizing on the existing facilitators can facilitate the development of robust antimicrobial stewardship initiatives essential to combating AMR in the nation. This study was conducted to explore the facilitators and barriers to ASPs' implementation and provide actionable insights to strengthen ASP and reduce AMR in the country.

## Methodology

### Study design and area

This study employed a qualitative, exploratory case study design to analyze the experience of healthcare providers on the implementation of Antimicrobial Stewardship Programs (ASPs) in Dar es Salaam's regional referral hospitals. The case study approach was selected for its capacity to provide a holistic, in-depth understanding of the processes involved in ASPs' implementation within distinct healthcare settings, emphasizing both contextual influences and stakeholder perspectives. The research was conducted in three regional referral hospitals in Dar es Salaam: Temeke, Amana, and Mwananyamala. They receive referrals from a minimum of 310 private and public hospitals, health centers and dispensaries with a capacity to provide outpatient service to about 1500,1200 and 1800 patients as well as a bed capacity of 304, 254 and 341 respectively (Hospital data). These hospitals play a critical role in the region's healthcare system, providing higher-level care and receiving referrals from lower-level facilities. Each hospital was capacitated and mandated to initiate ASP activities by the National Action Plan of 2017; this provided a relevant setting for examining program effectiveness and identifying site-specific facilitators and barriers. The hospitals operate under constraints typical of low- and middle-income countries (LMICs), including resource limitations and high patient volumes, significant factors when assessing the feasibility and efficiency of ASPs.

## Study Population and Sampling strategy

This study employed a qualitative design to explore perspectives of healthcare professionals directly involved in antimicrobial stewardship activities. The target population comprised healthcare workers engaged in antimicrobial prescribing, dispensing, administration, and microbiological diagnostics within hospital settings, as these roles are central to the implementation of Antimicrobial Stewardship Programs (ASPs)A purposive sampling strategy was employed to ensure the inclusion of participants with relevant expertise and firsthand experience. The initial sampling framework anticipated recruitment of approximately 20 participants to allow sufficient diversity of perspectives. However, recruitment was guided by the principle of data saturation rather than a fixed numerical target. The reduced sample did not compromise analytic rigor, as the interviews provided substantial depth and redundancy of information. Saturation was considered achieved when successive interviews yielded no new codes, themes, or conceptual insights, consistent with established qualitative methodological standards. Recruitment ceased at the point of saturation. Eleven [11] key informants (six males and five females) based on their professional roles and their direct involvement in ASP-related activities. This multidisciplinary approach was crucial for capturing a holistic view of ASPs' implementation, as each group contributes unique perspectives and faces distinct challenges. Out of the 11 participants, there were two doctors, one nurse, three pharmacists, three laboratory scientists, and two laboratory managers. Eligibility criteria included: [1] active employment in the selected hospitals, [2] direct involvement in antimicrobial prescribing, dispensing, administration, or diagnostic processes, and [3] at least one year of experience in antimicrobial stewardship-related activities to ensure adequate exposure to ASP implementation processes. Participants were purposively selected to capture variation in professional experience and career stage, enabling exploration of diverse perspectives within and across professional groups.

## Data collection techniques and procedure

Data collection occurred between April and May 2024 and involved a semi-structured interview (Supporting file labeled S1 Text). Individual key informant interviews were conducted with each participant to explore their experiences, perspectives, and perceived challenges and facilitators in ASP implementation. Participants were selected from the pre-formed ASP team at each hospital, all participants had experience of more than three years as medical personnel. This study did not include interns or volunteers from the selected hospitals. An interview guide was developed to ensure interview consistency while allowing flexibility to probe emergent themes. The guide included open-ended questions covering the following domains: 1) knowledge and understanding of ASP guidelines; 2) experiences with ASP; 3) facilitators of ASP implementation; 4) barriers to ASP implementation; and 5) suggestions for improvement. Probing questions were used to elicit detailed responses and explore specific issues that arose during the interviews. Each interview was conducted in a private setting within the hospital. Lastly, the minimum interview was 20 minutes, and the maximum was 33 minutes. All interviews were audio-recorded with the participant's consent to ensure accuracy in data capture

The interviews were collected in Kiswahili and English, enabling participants to express their thoughts in their preferred language and enhancing data richness. After obtaining the ethical clearance, pretesting of the tools was done at Amana hospital; three [3] participants who were not included in the study population were requested to pretest the tool.

## Data management and analysis

Data analysis was conducted using Braun and Clarke's (2006) thematic analysis approach. Audio recordings were verbatim transcribed to generate precise text documents for assessment. The accuracy of the transcripts was thoroughly checked against the original recordings through team reviews and discussions The analysis involved a manual process of coding, categorizing, and interpreting the data. First, transcripts were read multiple times to ensure familiarization. Each transcript was read, and meaningful text segments were assigned codes that captured their essence. Codes were descriptive and closely tied to the participants' own words. These codes were organized into broader

categories and then into overarching themes through an iterative and inductive process of review and refinement. Initially coding was performed in Kiswahili to maintain data integrity, the language in which most interviews were conducted. This approach helped retain contextual depth. Subsequently, translations were performed for report writing, ensuring that meaning and intent were preserved through a back-translation process where necessary to verify accuracy. Inductive coding was employed, allowing codes to be derived directly from the data rather than based on a pre-existing framework.

The clustered codes were examined to identify broader patterns and relationships, leading to the development of sub-themes and themes. Themes were refined by reviewing the data extracts and ensuring they accurately reflected the participants' narratives. Each theme received precise definitions and descriptive titles, making sure they were concise and appropriately represented the subject matter content.

Several strategies were used to ensure trustworthiness: transcripts were initially coded in the original language (Kiswahili) to preserve contextual depth, and analysis was an iterative process with regular team discussions to refine interpretations.

### Ethical approval and informed consent

Ethical approval for the study was obtained from the Muhimbili University of Health and Allied Sciences Institutional Review Board (IRB), on 18th March 2024 with reference number DA.282/298/01.C.2103. Informed consent was obtained from all participants before data collection, with assurances of confidentiality and the right to withdraw from the study at any time. To protect participant anonymity, personal identifiers were removed from transcripts, and all data were stored securely, with access limited to the research team.

## Results

The thematic analysis of the interview data revealed key insights into the factors influencing the implementation of Antimicrobial Stewardship Programs (ASPs) in the three regional referral hospitals. Two major themes emerged from the analysis: facilitators of ASP implementation and barriers to efficient implementation. These overarching themes were derived from several subthemes, providing a more granular understanding of the complexities surrounding ASP implementation and supported by illustrative quotes from the study participants (Table 1).

### Theme 1: Facilitators of ASP implementation in Dar es Salaam RRHs

**Institutional support for ASP activities.** Institutional leadership played a pivotal role in the success of ASPs. At one of the RRHs, strong leadership support was evident. The medical officer in charge was actively involved in ASP activities. Leaders advocated for necessary resources, prioritized ASP activities, and promoted staff engagement.

**Table 1. Summary of themes and sub-themes on facilitators and barriers to Antimicrobial Stewardship programs implementation in Dar es Salaam RRH.**

| Themes | Sub-themes |
| --- | --- |
| Facilitators of ASP implementation | Institutional Support for ASP Activities |
| | Capacity Building and Training |
| | Partnerships and External Support |
| | Role of regular monitoring of ASP activities |
| Barriers to ASP Implementation | Resource and Staffing Constraints |
| | Individual Barriers to Implementation |
| | Logistical Barriers to Implementation of ASP |

*"Our director is key to ensuring the ASP team gets the support it needs, and this has made a huge difference in our performance."* (Participant 08, Lab Scientist)

The support offered by hospital management in these hospitals has facilitated ASP teams to create initiatives that enhance diagnostic accuracies within the hospital. These included lowering the cost of culture and sensitivity tests so that most patients can afford accurate diagnoses since most patients attending these hospitals come from low socioeconomic statuses. This was highlighted by one of the participants mentioning that,

*"We reduced the cost of culture tests from Tsh 20,000/= to Tsh 10,000/=\*\*\*, making it easier for patients to afford, and now we do 2,500 to 3,000 tests annually. This has greatly improved diagnostic accuracy."* (Participant 09, Lab Scientist)

Nevertheless, at one of the hospitals, where leadership engagement was weaker, ASP struggled to gain ground in its initiation and implementation, and adherence to guidelines was inconsistent.

**Capacity building and training.** We found that ASP awareness varied significantly among participants. In two hospitals, healthcare workers demonstrated a stronger understanding of ASP protocols, owing to consistent training mostly done in-house.

*"We are trained to ensure the right use of antimicrobials, but the guidelines are not always followed across departments."* (participant 4, nurse)

These consistent in-house training sessions are conducted by ASP teams or external partners, equipping healthcare professionals with knowledge of ASP guidelines and reducing prescription errors. One of the pharmacists mentioned that,

**"***Before ASP, prescription errors were common, but now, with regular training and feedback, we see fewer incorrect doses being prescribed."* (Participant 4, Pharmacist)

(\*\*\* 1 US Dollar is equivalent to 2480 Tanzanian Shillings)

**Partnerships and external support.** Participant from one of the hospitals mentioned that external support from organizations such as Medicines, Technologies, and Pharmaceutical Services (MTaPS) and other implementing partners helped offset resource limitations by supplying laboratory reagents, educational materials, and diagnostic tools essential for ASP implementation."

*"MTaPS supported us with training and supplies, including laboratory commodities and antibiotics for testing. This collaboration made a big difference in our ASP."* (Participant 08, Lab Scientist).

**Role of regular monitoring of asp activities.** In two of the hospitals, feedback was delivered during weekly or quarterly meetings, where prescription errors and resistance patterns were discussed and adjustments were made to prescribing practices. These activities allow ASP teams to track antibiotic use, identify patterns in resistance, and provide corrective feedback to prescribers.

*"We do weekly therapeutic committee meetings where we give feedback on resistance levels. This helps raise awareness and make corrections in antibiotic use."* (Participant 09, Lab Scientist)

During these Continuous Medical Education (CME) meetings, which bring together specialists, physicians, pharmacists, and nurses who care for patients, ASP teams discuss pressing issues regarding antibiotic prescriptions in both

inpatient and outpatient settings, and implement quick corrections. However, it was noted that the feedback process can be slow, highlighting the need for more efficient M&E mechanisms.

"*Because we usually do CME meetings; if there is something we have seen that is hot and needs quick corrections, we usually present it at our CME meetings, which mostly reach many people.*" (Participant 03, Pharmacist)

**Barriers to efficient implementation of ASP Activities in Dar es Salaam RRHs**

**Resource and staffing constraints.** Resource limitations emerged as a significant barrier across all three hospitals. Participants consistently reported stockouts of essential diagnostic tools and delays in antibiotic procurement. These limitations hamper the ability of ASP teams to perform adequate surveillance and limit the effectiveness of stewardship interventions.

"*The Microbiology section often has challenges with stockouts; sometimes the sensitivity discs take a long time to be supplied because they are not available easily.*" (Participant 7, Lab Scientist)

Additionally, staff shortages also placed additional pressure on healthcare workers, who often had to prioritize immediate patient needs over ASP tasks due to high patient loads and limited personnel. These constraints hindered ASP teams' ability to conduct routine activities, such as weekly ward rounds and case audits.

"*AMS members are supposed to be on ward rounds every morning, but we only manage to do it on Friday due to time constraints. Also, the number of staff can be an issue; if one staff member goes on leave or s/he is transferred, there is a lot to do in the department, including routine activities.*" (Participant 08, Lab Scientist)

**Individual Barriers to Implementation.** Participants stated that healthcare workers, particularly prescribers, resist ASP implementation due to their level of experience at work, low awareness of the STGs, and doubts about the effectiveness of ASPs in AMR management. Furthermore, some participants said that prescribers prefer empirical treatments based on their experience rather than relying on guidelines or laboratory results.

"*Doctors still prescribe drugs based on experience rather than culture results. Some skip drugs that can be effective because they believe they no longer work.*" (Participant 01, Lab scientist)

These challenges extend beyond the healthcare system and workers to patients and relatives. The use of conventional manual methods in hospitals leads to a four- to seven-day turnaround time (TAT) for culture samples, causing patients to request antibiotics and never follow up on culture results.

"*Patients themselves find it difficult to wait for the results; the turnaround time (TAT) for culture and sensitivity is a bit longer than other tests, so patients sometimes feel it's difficult to wait.*" (Participant 9, Lab scientist)

**Logistical barriers to implementation of ASP activities.** It was found that delays in procurement, inadequate supply chains, and the lack of automated diagnostic systems hinder the efficient implementation of ASPs. All three hospitals depended on a central procurement organ, the Medical Stores Department (MSD), which received and distributed to all government healthcare facilities. This emerged as a challenge because it sometimes takes long to order consumables and equipment from MSD. These challenges make it difficult to maintain a consistent supply of antibiotics, diagnostic

tools, and other resources essential for ASP operations. As a result, hospitals struggle to maintain continuous monitoring and reporting activities.

*"Challenges often involve procurement and access to supplies like microbiology tools and culture media… because we in government hospitals depend on MSD, and you find that MSD doesn't have those discs or it takes a while to get them, which also leads to having fewer drugs; you cannot do good sensitivity; you can't prepare good antibiogram; so that's the challenge we face with procurement." (Participant 01, Lab Scientist)*

## Discussion

The study found that awareness of ASP guidelines serves as the foundation for effective stewardship, yet this awareness is inconsistent among healthcare workers. While some staff members are well-informed, others lack a deep understanding of ASP protocols, often because current training efforts do not reach all personnel or because stewardship is not prioritized in daily practice.This gap mirrors findings in other East African contexts, where irregular training and poor dissemination of guidelines have led to awareness inconsistencies [12].Additionally, adherence to Standard Treatment Guidelines (STGs) varies throughout departments, even if the enforcement of ASPs has lessened the prior excessive reliance on broad-spectrum antibiotics. A lack of ongoing monitoring or the belief that these rules are excessively tight for demanding clinical settings may be the cause of non-compliance in some places.

Leadership engagement emerged as a decisive factor in ASP implementation. Two out of the three hospitals with institutional support characterized by strong management involvement, facilitative supervision, and willingness to allocate internal resources demonstrated better stewardship performance. Supportive leadership environments enabled more consistent monitoring, improved coordination between pharmacists, clinicians, and laboratory personnel, and strengthened accountability mechanisms. This finding aligns with research emphasizing leadership commitment as a core determinant of successful ASP implementation, especially in resource-constrained settings [20,24]. The same is true in developed countries. Effective ASP implementation is not solely dependent on resources but relies heavily on a supportive organizational culture fostered by engaged leadership. In the USA, ASP success is closely tied to the involvement of hospital management, and leadership has been shown to significantly enhance the sustainability of ASP initiatives [25], a notable improvement in adherence to guidelines and a reduction in antibiotic misuse [26]. The consistency of this finding across different healthcare settings underscores the universal importance of leadership in driving ASP initiatives. Therefore, leadership engagement should be prioritized as a core component of ASP implementation strategies.

Investing in comprehensive and ongoing training programs is essential to ensure that healthcare workers have the knowledge and skills necessary to implement ASPs effectively. In this study, weekly or quarterly feedback meetings facilitated peer learning and enabled ASP teams to address emerging patterns in resistance and prescribing errors. However, the timeliness and reach of feedback varied, with some hospitals relying heavily on periodic CME sessions that did not effectively engage all departments. The disparity in awareness and adherence to ASP guidelines between hospitals highlights the need for more consistent training programs [27]. Inadequate education and training among healthcare professionals regarding antimicrobial stewardship principles, particularly in Kenya, Vietnam, and Indonesia, contribute to inconsistent adherence across departments and suboptimal prescribing practices, as highlighted in literature [27–29] Regular training and feedback mechanisms should be established to reinforce ASP principles and best practices.

External support from organizations and partnerships helped to offset resource limitations by providing essential supplies and training for ASP implementation. Leveraging partnerships and seeking external support can be a valuable strategy for overcoming resource barriers and enhancing ASP capacity. This highlights the critical role that external collaborations can play in facilitating ASP implementation in resource-limited settings. Several studies in LMICs have highlighted the importance of external support for healthcare initiatives. For instance, a study in Nepal emphasized that partnerships were crucial in providing resources and technical expertise for ASP implementation, similar to our finding [20]. In contrast,

while our study focused on the direct benefits of material support, other research has also pointed to the role of partnerships in facilitating knowledge transfer and capacity building, which can have a longer-term impact on ASP sustainability [30–32]. The similarity underscores the reliance of many LMICs on external aid to supplement limited domestic resources for essential healthcare programs like ASPs. The difference suggests that while material support is crucial in the short term, building local capacity through collaborative knowledge exchange is vital for long-term ASP success.

Additionally, regular monitoring of ASP activities, including feedback sessions and committee meetings, facilitated the tracking of antibiotic use, identifying resistance patterns, and correcting prescribing practices. The CDC core elements for ASP advocate for regularly monitoring antibiotic use and resistance patterns [2]. These guidelines are crucial for tailoring interventions to local needs [33]. A study in China found a significant reduction in antibiotic consumption and resistance following a structured antibiotic use policy, emphasizing the importance of following established guidelines [34]. Antibiograms offer real-time data on prescription patterns and resistance trends; on the other hand, feedback sessions have been shown to increase adherence to ASP standards [35]. Mushtaque et al. report that compliance and patient outcomes can be further enhanced by incorporating feedback into routine procedures [36]. Establishing robust monitoring and evaluation mechanisms is essential for ensuring the effectiveness and sustainability of ASPs.

In contrast, multiple barriers limited the full operationalization of ASPs, particularly in the hospital where the program had not yet been implemented. Resource and funding constraints were a major impediment, affecting procurement of diagnostic supplies, preventing routine antibiogram development, and leading to frequent stockouts of essential laboratory materials and antibiotics. The absence of automated diagnostic systems also prolonged turnaround time for culture results, reducing clinicians' willingness to wait for laboratory confirmation and reinforcing empirical treatment habit, Unlike high-income countries, where ASP teams are well-established, LMICs face human resource constraints [37]. This shortage hinders ASP implementation and underscores the need for capacity-building initiatives, such as continuous medical education and mentorship programs [38]. The recurrence of this issue in various LMIC contexts highlights the systemic challenges these countries face in establishing sustainable interventions and programs [12].

Workload pressures further undermined ASP engagement. Many healthcare workers were already stretched by heavy patient loads, leaving limited time for stewardship duties such as documentation, ward rounds, or audit participation.

Furthermore, resistance from healthcare workers, particularly prescribers, due to experience-based practices and skepticism about ASP effectiveness poses a challenge to ASP implementation. Studies in Nigeria and South Africa have also documented physician reluctance due to concerns over clinical autonomy [32,39]. This results in frequent empirical antibiotic use without culture and sensitivity testing. Similar trends are reported in Ghana and Zambia, where diagnostic limitations contribute to inappropriate prescribing [5,40]. This emphasizes the importance of addressing behavioral and attitude issues among healthcare professionals to promote an antimicrobial stewardship culture.

Lastly, logistical barriers hindered ASP implementation were identified as inefficient procurement processes and the lack of automated diagnostic systems. By interfering with the availability of necessary resources and postponing diagnostic procedures, which are crucial for efficient antimicrobial stewardship, these logistical obstacles jeopardize ASP efforts [41]. Delays in procurement and inadequate supply chains have been reported in other LMICs, highlighting systemic challenges in healthcare logistics [15,19,40]. To ensure timely resource availability and streamline ASP operations, it is vital to invest in diagnostic infrastructure and improve procurement mechanisms like barcoding and stock management software or establishing a system of forecasting demand for key ASP supplies.

## Conclusion

Implementing Antimicrobial Stewardship Programs (ASPs) in Dar es Salaam's regional referral hospitals have demonstrated significant potential in reducing antimicrobial resistance and improving patient care outcomes. The efficiency and sustainability of these programs are fundamentally challenged by systemic resource limitations and behavioral resistance. The key conclusion is that despite the recognized benefits, financial constraints were a significant barrier to

ASP implementation, leading to a detrimental reliance on external funding that limits long-term success. Chronic issues inherent to LMIC settings including staff shortages and high workload, logistical barriers such as delayed procurement of diagnostic supplies, and individual reluctance to change among prescribers, severely impede the consistent and efficient operation of ASPs across all three study sites. Addressing the limitations identified in this study, particularly around diagnostic capacities and adherence consistency, is essential to enhance ASP implementation.

### Recommendation

These findings underscore several implications for ASP policy and practice in Tanzania and similar LMICs. Hospitals must integrate ASP activities with other programs, particularly Infection Prevention and Control (IPC), to ensure a coordinated approach. Furthermore, healthcare facilities need to develop and implement specific performance indicators and regular reporting systems to track ASP progress, identify gaps, and enhance communication and coordination among multidisciplinary stakeholders. Concurrently, the National AMR Multisectoral Coordination Committee should prioritize ASP management, provide clear directives, and guide facilities in establishing robust surveillance programs to monitor antibiotic use and resistance patterns.

### Study limitations and mitigation

This study faced several methodological limitations. Although the National Action Plan recommends that each hospital have at least six members in its antimicrobial stewardship (ASP) team, the actual team composition varied considerably across the three regional referral hospitals, resulting in a smaller-than-anticipated participant pool.

Nevertheless, considering the qualitative character of the research and the emphasis on in-depth experience analysis, this sample size is deemed appropriate [42]. Interviews were conducted in both Kiswahili and English, and despite careful transcription and cross-checking, some risk of translation bias remains, as certain linguistic nuances may not have been fully captured. Additionally, the study was limited to regional referral hospitals in Dar es Salaam, potentially restricting the generalizability of the findings. To mitigate these limitations, diverse participants from different hospital departments were included to ensure a broad perspective on ASP implementation. While these factors may have shaped the depth and scope of the data, the participants involved held key roles in stewardship activities and provided rich insights that strengthen the study's analytic value. Future research should consider expanding to other regions and incorporating mixed-method approaches to provide a more comprehensive evaluation of ASP efficiency.

### Supporting information

**S1 Text. Key Informant Interview guide.**
(PDF)

**S1 Data. Transcript files.**
(ZIP)

**S2 Data. Transcript files.**
(ZIP)

### Author contributions

**Conceptualization:** Berthania Magesa, Nathanael Sirili.

**Data curation:** Berthania Magesa.

**Formal analysis:** Berthania Magesa.

**Funding acquisition:** Berthania Magesa.

**Investigation:** Berthania Magesa, Margareth Mhame.

**Methodology:** Berthania Magesa, Luco Mwelange, Nathanael Sirili.

**Project administration:** Berthania Magesa, Margareth Mhame, Luco Mwelange.

**Resources:** Berthania Magesa.

**Software:** Berthania Magesa.

**Supervision:** Luco Mwelange, Nathanael Sirili.

**Validation:** Berthania Magesa, Nathanael Sirili.

**Visualization:** Berthania Magesa, Margareth Mhame, Nathanael Sirili.

**Writing – original draft:** Berthania Magesa.

**Writing – review & editing:** Berthania Magesa, Margareth Mhame, Luco Mwelange, Nathanael Sirili.

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
