## [Decision Letter · Decision Letter 0]

30 Sep 2025

PGPH-D-25-01608

Barriers and facilitators for the implementation of Antimicrobials Stewardship Programs (ASP)  in Dar es Salaam Regional Referral Hospitals (RRHs)

Dear Dr. Magesa,

Thank you for submitting your manuscript to PLOS Global Public Health. After careful consideration, we feel that it has merit but does not fully meet PLOS Global Public Health’s publication criteria as it currently stands. Therefore, we invite you to submit a revised version of the manuscript that addresses the points raised during the review process.

We look forward to receiving your revised manuscript.

Kind regards,

Emmanuel Olamijuwon, Ph.D

Academic Editor

Journal Requirements:

1. We note that your Data Availability Statement is currently as follows: “The data is provided as part of the submitted article.”

Please confirm at this time whether or not your submission contains all raw data required to replicate the results of your study. Authors must share the “minimal data set” for their submission. PLOS defines the minimal data set to consist of the data required to replicate all study findings reported in the article, as well as related metadata and methods (https://journals.plos.org/globalpublichealth/s/data-availability#loc-minimal-data-set-definition).

If your submission does not contain these data, please either upload them as Supporting Information files or deposit them to a stable, public repository and provide us with the relevant URLs, DOIs, or accession numbers. For a list of recommended repositories, please see https://journals.plos.org/globalpublichealth/s/recommended-repositories.

2. Please ensure that the Title in your manuscript and the Title in your online submission form are the same.

Additional Editor Comments (if provided):

Reviewers' comments:

Reviewer's Responses to Questions

**Comments to the Author**

1. Does this manuscript meet PLOS Global Public Health’s publication criteria?

Reviewer #1: Yes

Reviewer #2: Partly

Reviewer #3: Yes

Reviewer #4: Yes

Reviewer #5: Yes

Reviewer #6: Yes

2. Has the statistical analysis been performed appropriately and rigorously?

Reviewer #1: N/A

Reviewer #2: Yes

Reviewer #3: Yes

Reviewer #4: N/A

Reviewer #5: No

Reviewer #6: Yes

3. Have the authors made all data underlying the findings in their manuscript fully available (please refer to the Data Availability Statement at the start of the manuscript PDF file)?

Reviewer #1: Yes

Reviewer #2: No

Reviewer #3: No

Reviewer #4: Yes

Reviewer #5: Yes

Reviewer #6: Yes

4. Is the manuscript presented in an intelligible fashion and written in standard English?

Reviewer #1: Yes

Reviewer #2: Yes

Reviewer #3: Yes

Reviewer #4: Yes

Reviewer #5: Yes

Reviewer #6: Yes

Reviewer #1: Comments

The present study explored the experience of health care providers to assess the barriers and facilitators of Antimicrobial stewardship program in Tanzania.

Abstract

Line 29 to 30; The sentence “Significant 30 barriers included” is a hanging statement that is not complete

Methods

Line 126: “Strategy” should be added to the heading of “Study Population and Sampling”

Line 131: The “S” in six should be small letter

Line 184: Kindly indicate the date of ethical approval was obtained

Results

Line 204: The Title of the table is too short. The authors should make the title more informative

Line 204: I also suggest the column for the theme should come first before the column for sub-themes

Line 219: The sentence “We reduced the cost of culture tests from 20,000/= to 10,000/=”. Kindly indicate the currency of the denominator. This maybe converted to USD

Line 221: Be consistent the information in bracket was italicized but in the previous line 213, it was not italicized. Same for lines 232,237, 253

Line 241: After partners, there should be a full stop before These

Discussion

Line 378: The sentence “without culture testing” should be paraphrase as “without culture and sensitivity testing”

Reviewer #2: Overall, the manuscript titled the barriers and facilitators to implementing antimicrobial stewardship programs (ASPs) in regional referral hospitals in Tanzania, addresses an important public health issue especially in settings where ASP implementation faces unique challenges. The strengths of this paper include the following:

1. The manuscript has clearly outlined the research objective and appropriate use of qualitative methodology (thematic analysis).

2. It also utilised multidisciplinary cadres (doctors, nurses, pharmacists, laboratory staff) thereby providing diverse perspectives.

3. The results are well-structured into facilitators and barriers of ASP and they are supported with illustrative participant quotes.

4. The manuscript’s discussion connects findings to existing literature at global level and sub-Saharan level which strengthens relevance.

However, it also has a few areas of improvements which if addressed can strengthen the paper. The following are some of the areas that I have identified that needs to be addressed.

1. In the abstract (Lines 15–19), the use of the word ‘efficiency’ is misleading for this paper because the study is not measuring efficiency but rather implementation experiences.

2. In the introduction, line 59-62, the paper talks about National AMR data systems that remain underdeveloped and efforts from the East Africa Public Health Laboratory Network (EAPHLN) are attempting to address these deficits. But it is not referenced for us to truly verify if these efforts are being done. A reference is needed.

3. Methods: the inclusion and exclusion criteria is not clearly defined and it is not clear if the paper only included senior staff or they also included junior staff to participate.

4. Methods: Line 128–137: the study only included 11 participants but it has not justified as to whether data saturation was reached after 11 interviews. I suggest that they clearly indicate if saturation was reached

5. Methods: Line 148 talks about a data collection tool but it not attached, I suggest that they include supplementary file for transparency

6. Results: Line 193–199 introduce themes and sub-themes with a very good structure but with only 11 participants, the generalizability of this study is very limited. I would suggest including it in the limitation section (which should be indicated) because this is a strong limitation and the findings may not be generalised beyond the study settings

7. Results: Line 204, Table summarises facilitators and barriers, but the paper could be strengthened by indicating the number of participants who supported each sub-theme (e.g., 7 out of 11 participants mentioned stockouts).

8. Discussion: The manuscript lacks a clear limitations section which can bring out critical limitation like the small sample size, potential interviewer and translation bias.

Reviewer #3: General comments and some key concerns:

There are some grammar issues especially in the use of the tenses. There are some terms that need to be explained including antimicrobial agent, antibacterial agent, antibiotic, medicines and drugs, misuse and inappropriate use etc.

1. Abstract

The authors need to be align the objectives to the title (line 17). In the methods (line 20), the sentence is not clear and it needs to be paraphrased. In line 29-30, the sentence is incomplete. In the conclusion, most of the findings would be in the results. And what were the key conclusions from the study based on the findings of the study.

2. Introduction

Line 56-57, the sentence need citation. The authors need to highlight the core concepts of ASP and the reported challenges in implementing the ASP in healthcare facilities. In line 93-94, what is the national prevalence of AMR in the country if any? The authors should highlight to what extent, the ASP is implemented in those hospitals and nationally.

3. Methods

The authors should write “Methodology” as a sub-title for this section instead of “Materials and Methods”. The authors should elaborate more on the three regional referral hospitals in Dar es Salaam: Amana, Mwananyamala, and Temek in terms of their location and capacity. In the study population and sampling (line 126-138), 11 key informants (Six males and five females) Doctors, nurses, pharmacists, and laboratory staff were included. However, the issue is how many were males and females of these different cadres (doctors, nurses, pharmacist and laboratory staff), at the same level of training and different career stages and with work experience or have been exposed to ASP training? This may have created some bias and hence influence the outcomes. Why were administrators or managers of the hospitals not included in the study?

4. Results

Line 240-241, there is grammar issue in the sentence. Of the selected hospital, were they experiencing the same extent of barriers and facilitators and if not, to what extent were these in the selected study facilities?

5. Discussion

The authors should indicate the major limitations of the study.

6. Conclusion

The conclusions should be based on the findings of the study which is not the case for the present study.

Reviewer #4: This manuscript endeavors to address the barriers and facilitators of Antimicrobial Stewardship Program (ASP) implementation in three regional referral hospitals in Dar es Salaam, Tanzania. The study was based on exploratory qualitative case study design, and it is of public health importance. The authors conducted 11 semi-structured interviews with healthcare providers and analyzed the data thematically. Findings highlight leadership, training, partnerships, and monitoring as key facilitators, while resource limitations, staffing shortages, individual resistance, and logistical barriers impede ASP implementation. The study is timely and relevant for low- and middle-income countries (LMIC) settings where antimicrobial resistance (AMR) is a major challenge.

Abstract

Well written

Introduction

Well written

Line 50, 54 etc: citation in-text should have space from text.

There is a minor redundancy between global AMR threat and local Tanzanian challenges. This can be linked to strengthening focus.

Methods

Well written and structured

The only limitation could be that 11 participants across 3 hospitals is small, which may lead to the risk of limited representativeness. The authors should justify why such a small study population and sampling.

Results

Fairly well written, however, some quotes are repetitive which I would suggest condensing.

Additionally, there is no indication of variation across the three hospitals (Amana, Mwananyamala, Temeke). This would strengthen contextual understanding of the study.

Discussion

Well written. However, it may be of benefit, if it is not too descriptive (summarizing results again) rather than critically analyzing.

Conclusion

It is clearly written and aligned with objectives.

Reviewer #5: Summary

This manuscript addresses barriers and facilitators to implementing Antimicrobial Stewardship Programs (ASPs) in three regional referral hospitals in Dar es Salaam, Tanzania. The authors use an exploratory qualitative design with interviews of healthcare workers to capture their perspectives. The topic is timely given the global urgency of antimicrobial resistance (AMR), and the focus on a low-resource setting makes the work important for PLOS Global Public Health.

Major Comments

1. Significance: The study adds useful insights into ASP implementation in LMICs. However, the discussion could be strengthened by placing the findings in a broader regional/global context. Comparing these results with similar experiences in other African or Asian countries would help readers judge how generalizable the findings are.

2. Methods: The paper would benefit from more detail about the coding process and how themes were developed. For instance, how was consistency ensured between coders, and how was saturation determined with only 11 participants? Adding a short description of the analytic steps (or even an appendix with a coding tree) would improve transparency.

3. Policy/Practice Relevance: The conclusions highlight the need for resources and leadership, but they remain general. It would be stronger if the authors suggested specific actions for hospitals or policymakers (e.g., integrating ASP indicators into routine hospital reporting, improving supply chain management, or including ASP metrics in performance evaluations).

4. Limitations: While the small sample size is acceptable for qualitative work, it should be more explicitly discussed as a limitation. In particular, the reader needs reassurance that the findings are not biased toward only a few professional perspectives

Minor Comments

• The abstract repeats some phrases between the background and objectives — this could be tightened.

• The presentation of barriers and facilitators could be more structured, e.g., grouped into “system-level,” “institutional,” and “individual” factors for clarity.

• Some sentences in the discussion are long and could be shortened for readability.

• Figures/tables: the summary table of themes is useful but could be reformatted for easier interpretation.

Overall Recommendation

The manuscript tackles an important problem in global health and provides relevant qualitative insights. With more detail on methods and a clearer, action-oriented conclusion, this paper could make a valuable contribution to the literature on ASPs in LMICs.

Reviewer #6: INTRODUCTION

Summary: The introduction gives a strong rationale for AMR and Antimicrobial Stewardship Programs (ASPs), contextualized from global to regional to national (Tanzania). It cites global burden data and includes local resistance rates, which is a strength. However, the section suffers from repetition (ASPs defined multiple times), unclear transitions, speculative phrasing, and omission of key global frameworks (GLASS, AWaRe, One Health). These gaps reduce clarity and weaken alignment with international AMR discourse.

Major Points to Address:

- M1: ASPs are defined or described multiple times, including at the very beginning and again in mid-introduction. Keep one concise definition at the start. Later mentions should expand on ASP components rather than redefine them.

--Examples: L1 - “Antimicrobial stewardship programs (ASP) are coordinated efforts…” L79 - Antimicrobial Stewardship Programs (ASPs) are among the pivotal strategies…

Suggestion

- M2: Global frameworks such as GLASS, AWaRe, and One Health framing are not referenced when discussing surveillance gaps. Including these will support alignment with global AMR standards, which will strengthen the claims. Mention GLASS as WHO’s surveillance framework.", "Clarify whether Tanzania reports to GLASS or uses AWaRe classification.

-- Examples: L59-L61: National AMR data systems remain underdeveloped, with efforts like the East Africa Public Health Laboratory Network (EAPHLN) indicating ongoing attempts to address these deficits.

-- Suggestion for rewrite: Surveillance remains limited; while WHO’s GLASS provides a global framework, participation from Tanzania is incomplete, and AWaRe-classified antibiotic use is not yet consistently tracked."

- M3: The phrase in L82-83: ASPs aim to prolong the effective life of antibiotics, improve their utilization, and research novel antibiotics, is speculative and suggests the ASPs take part in antibiotic research which misrepresents their role and the reference provided does not support this claim. I would suggest to limit ASP scope to stewardship activities (optimizing use, reducing misuse) OR rephrase or remove the claim about antibiotic research.

- M4: L85-86: The transition from regional AMR context to Tanzania-specific context is abrupt. Readers may struggle to follow the narrowing of scope. Add a bridging sentence narrowing from Sub-Saharan Africa → East Africa → Tanzania. Clarify why Tanzania is a focus case.

-- Suggestion: "Within this regional context, Tanzania provides a salient case, having implemented a National Action Plan on AMR since 2017 with specific ASP activities."

Minor Points:

m1: L50: Already typed out in L48. General comment, make sure the abbreviations are consistent throughout.

m2: L93 – L97: I suggest alongside citations explicitly state the sources of resistance data, are they from national / international or subnational levels?

m3: L51-54: Avoid colloquial phrasing. Rephrase please. This is a general comment. “make common infections untreatable” is strong and a bit conversational in tone, try “render infections increasingly difficult to treat”

METHODS:

Summary: The Methods section describes a qualitative case study of ASP implementation in three referral hospitals in Dar es Salaam. Strengths include a clear statement of study design, purposive sampling rationale, and detailed data collection and analysis procedures. Ethical approval is explicitly stated. However, the section lacks important methodological details required for rigor and transparency: COREQ items such as reflexivity (researcher role, positionality), intercoder reliability, saturation criteria, and audit trail are missing. The sampling description is inconsistent (laboratory staff vs. laboratory managers), and justification for sample size (11 participants) is absent. Data collection details are present but lack evidence of instrument piloting or validation. Analysis description is broad and descriptive but does not detail coding team composition or measures to enhance reliability beyond team discussions. These gaps limit reproducibility and may undermine credibility of the findings.

Major Issues:

-M1: L130: The study includes 11 participants but provides no rationale for why this number was sufficient to achieve thematic saturation. I would add an explicit rationale for the sample size (e.g. expected saturation, diversity of roles) and I would include a sentence, stating whether saturatiın was achieved and how it wasa assessed.

-- Suggestion: A total of 11 participants were recruited; this number was determined based on the expectation of reaching thematic saturation across key stakeholder groups.

M2: L132: Earlier text says 'laboratory staff,' but later specifies 'three laboratory scientists, three laboratory managers,' which adds confusion. Ensure participant categories are consistent throughout. Clarify whether laboratory staff were scientists, managers, or both.

-- Suggestion: The sample included three pharmacists, three laboratory scientists, and two laboratory managers (collectively referred to as 'laboratory staff').

- M3: L162: There is no description of the research team, their professional background, training, or positionality. Reflexivity is critical in qualitative research for transparency and bias reduction (COREQ requirement).Describe the research team roles (interviewers, analysts) and their relationship to participants. Reflect on how researcher background may have influenced data collection and analysis.

M4: L181-183: Trustworthiness is mentioned generally, but key elements like intercoder reliability, audit trail, or triangulation are not detailed. Lack of detail undermines rigor and reproducibility. pecify whether multiple coders were used and how disagreements were resolved. Describe whether an audit trail or triangulation was applied

minor issues:

m1: L154: Provide the actual range (minimum-maximum) of interview durations instead of approximates.

m2: L148: Please inform whether the interview guide was piloted or validated before.

m3: In general I would state whether any interviews were excluded or incomplete.

RESULTS:

Summary: The Results section presents a clear thematic analysis structured into facilitators and barriers of ASP implementation in Dar es Salaam referral hospitals. Strengths include the use of direct participant quotes, identification of multiple subthemes (institutional support, capacity building, external partnerships, monitoring, resource constraints, individual and logistical barriers), and contextual examples such as reduced costs for culture tests. However, reporting is largely descriptive and lacks methodological rigor: traceability between quotes and themes is inconsistently demonstrated, no evidence of saturation is reported, participant roles are not always clear in quotes. Greater precision and adherence to COREQ standards (quote–theme linkage, transparency of coding) are needed.

Major Points:

M1:Interpretation creeps into Results. Example: L224: “indicating the importance of supportive leadership system in implementing ASP”. This phrase interprets the findings instead of simply reporting what participants said. Results should focus on participant data; interpretation belongs in Discussion.

Suggestion for rewrite: At one of the hospitals, where leadership engagement was weaker, ASP struggled to gain ground in its initiation and implementation, and adherence to guidelines was inconsistent.

M2: Traceability of quotes to themes. Example: L199: “supported by illustrative quotes from the study participants (table 1).”

While quotes are provided, the connection between subthemes and participant voices is not systematically demonstrated. COREQ requires transparent linkage between themes and supporting data. Explicitly show how each quote maps to a subtheme and ensure all subthemes are backed by at least one direct quote.

M3: Participant identification lacks clarity. Participant idnetifiesers are inconsistently reported (“Lab” vs “Lab Scientist” vs. “Pharmacist”).

Minor Issues:

m1: L196-197: Capitalize consistently. Facilitators of ASP Implementation' and 'Barriers to ASP Implementation'.

DISCUSSION AND CONCLUSION:

Global Summary: The Discussion and Conclusion effectively summarize key facilitators (leadership, training, partnerships, monitoring) and barriers (resource gaps, staffing shortages, prescriber resistance, logistical challenges) to ASP implementation in Dar es Salaam hospitals. Strengths include linking findings to literature from LMICs and HICs, highlighting parallels and contrasts, and referencing CDC core elements. However, the section suffers from several issues: interpretive overreach in places, missing integration of global frameworks (GLASS, AWaRe, One Health), lack of explicit acknowledgement of study limitations, and vague claims about ASPs 'reducing antimicrobial resistance' without outcome data. The Conclusion repeats the Discussion rather than synthesizing implications. Greater precision and cautious phrasing are needed.

Major Points of Revision:

M1: In the Conclusion: L393-395: The phrase: “ demonstrated significant potential in reducing antimicrobial resistance and improving patient care outcomes” risks overreaching the data, as the study did not measure resistance rates or patient outcomes. Rephrase to reflect qualitative scope and limit to reporting perceived or anticipated benefits, not measured outcomes.

M2: In the Discussion: L357: The Discussion cites CDC guidance but omits WHO GLASS, AWaRe, or One Health framing when contextualizing findings.

M3: Study limitations are missing. No explicit discussion of limitations (sample size, generalizability, self-report bias, language/translation issues). Add a dedicated paragraph acknowledging limitations. Discuss implications for transferability of findings.

Minor Points of Revision:

m1: L321-322: This is a sentence fragment, either delete or incorporate to prior or latter sentence.

m2: L317: typo in challenge,s. Delete the comma (,).

m3: I think adding a statement whether the research findings are transferable beyond Dar es Salaam would work nicely.

m4: Streamline redundancy in Conclusion and Discussion. L397: This repeates content from the Discussion, streamline to one concise synthesis.

**Do you want your identity to be public for this peer review?** For information about this choice, including consent withdrawal, please see our Privacy Policy

Reviewer #1: **Yes:** Abdurrahman Hassan Hassan

Reviewer #2: **Yes:** Hope Kalasa

Reviewer #3: No

Reviewer #4: No

Reviewer #5: **Yes:** Emmanuel Appiah-Kubi

Reviewer #6: **Yes:** Ege Dedeoğlu

---

## [Decision Letter · Decision Letter 1]

18 Jan 2026

PGPH-D-25-01608R1

Barriers and facilitators for the implementation of Antimicrobials Stewardship Programs (ASP)  in Dar es Salaam Regional Referral Hospitals (RRHs)

Dear Dr. Magesa,

Thank you for submitting your manuscript to PLOS Global Public Health. After careful consideration, we feel that it has merit but does not fully meet PLOS Global Public Health’s publication criteria as it currently stands. Therefore, we invite you to submit a revised version of the manuscript that addresses the points raised during the review process.

We look forward to receiving your revised manuscript.

Kind regards,

Syed Masud Ahmed, MD, PhD

Academic Editor

Journal Requirements:

Additional Editor Comments (if provided):

Reviewers' comments:

Reviewer's Responses to Questions

**Comments to the Author**

Reviewer #1: All comments have been addressed

Reviewer #4: All comments have been addressed

Reviewer #5: All comments have been addressed

Reviewer #6: All comments have been addressed

publication criteria?

Reviewer #1: Partly

Reviewer #4: Yes

Reviewer #5: Yes

Reviewer #6: Yes

3. Has the statistical analysis been performed appropriately and rigorously?

Reviewer #1: N/A

Reviewer #4: N/A

Reviewer #5: Yes

Reviewer #6: Yes

4. Have the authors made all data underlying the findings in their manuscript fully available (please refer to the Data Availability Statement at the start of the manuscript PDF file)?

Reviewer #1: Yes

Reviewer #4: Yes

Reviewer #5: Yes

Reviewer #6: Yes

5. Is the manuscript presented in an intelligible fashion and written in standard English?

Reviewer #1: Yes

Reviewer #4: Yes

Reviewer #5: Yes

Reviewer #6: Yes

Reviewer #1: Comment

Highly relevant topic aligned with global AMR priorities. Appropriate qualitative design and clear use of Braun & Clarke thematic analysis.

Introduction

Terminology consistency, the use of ASP and ASPs. Line 47 ASP was used, but line 72 ASPs was used

Methods

Also considering the fact that the number of the subjects or respondents are not much, that might affect the general conclusion of these findings. 11 purposively selected healthcare providers

Line 136 to 141: The explanation of the study population was narrated like a result . Please paraphrase these sentences to make it sound methodological

Line 187. Please insert date ethical approval; was obtained

Result

Line 199, “table 1”. “T ” should be capital letter

Line 204: The title of the table is too short. Kindly provide more descriptive narration of the table

Line 241: Full stop after partners.

Line 254: full meaning of CME

Line 269: disks, it should be discs

Line 287: “(Participant 01, Lab)” complete the Lab ……. scientist or personnel?

Reviewer #4: The authors have adequately responded to the reviewers' comments, and the quality of the manuscript has improved. I therefore have no further comments.

Reviewer #5: I have reviewed the revised manuscript and confirm that the authors have adequately addressed all comments raised in my initial review. The Discussion section has been appropriately revised to reduce descriptive repetition and provide stronger critical analysis in line with international literature. The recommendations have been strengthened with clearer, actionable policy and practice implications, and the abstract has been improved for clarity and conciseness.

I have no further comments and support the manuscript for publication.

Reviewer #6: No further comments.

**Do you want your identity to be public for this peer review?** For information about this choice, including consent withdrawal, please see our Privacy Policy

Reviewer #1: **Yes:** Abdurrahman Hassan Jibril

Reviewer #4: No

Reviewer #5: **Yes:** EMMANUEL APPIAH-KUBI

Reviewer #6: No

---

## [Editor Report · Decision Letter 2]

2 Mar 2026

Barriers and facilitators for the implementation of Antimicrobials Stewardship Programs (ASP)  in Dar es Salaam Regional Referral Hospitals (RRHs)

PGPH-D-25-01608R2

Dear Ms Magesa,

We are pleased to inform you that your manuscript 'Barriers and facilitators for the implementation of Antimicrobials Stewardship Programs (ASP)  in Dar es Salaam Regional Referral Hospitals (RRHs)' has been provisionally accepted for publication in PLOS Global Public Health.

Best regards,

Syed Masud Ahmed, MD, PhD

Academic Editor